# BiodivOnto: Towards a Core Ontology for Biodiversity

Nora Abdelmageed[1,2,3] (iD), Alsayed Algergawy[1] (iD),

Sheeba Samuel[1,3] (iD), and Birgitta König-Ries[1,3] (iD)

[1] Heinz Nixdorf Chair for Distributed Information Systems
[2] Computer Vision Group
[3] Michael Stifel Center Jena
Friedrich Schiller University Jena, Germany
`firstname.lastname@uni-jena.de`

**Abstract.** Biodiversity is the variety of life on earth which covers the evolutionary, ecological, and cultural processes that sustain life. Therefore, it is important to understand where biodiversity is, how it is changing over space and time, the driving factors of these changes and the resulting consequences on the diversity of life. To do so, it is necessary to describe and integrate the conditions and measures of biodiversity to fully capture the domain. In this paper, we present the design of a core ontology for biodiversity aiming to establish a link between the foundational and domain-specific ontologies. The proposed ontology is designed using the fusion/merge strategy by reusing existing ontologies and it is guided by data from several resources in the biodiversity domain.

**Keywords:** Biodiversity· Knowledge Representation· Core Ontology.

## 1 Introduction

The recent IPBES global assessment[4] foresees a dramatic decline in biodiversity and caused by this a dramatic decline in important ecosystem functions. To preserve biodiversity, research to understand its underlying mechanisms is needed which requires integrated data [6]. An increasing amount of heterogeneous data is generated and publicly shared in biodiversity research. There are also a lot of efforts to semantically describe biodiversity datasets and research outputs. Multiple ontologies, like ENVO[5] and IOBC[6], model specific parts of the domain. However, in order to support integrative biodiversity research, there is a growing need to bridge between the more refined biodiversity concepts and general concepts provided by the foundational ontologies.

   Core ontologies provide a precise definition of structural knowledge in a specific field that connects different application domains  [3,4,10]. They are located

---

[4] `https://ipbes.net/global-assessment`
[5] `https://bioportal.bioontology.org/ontologies/ENVO`
[6] `https://bioportal.bioontology.org/ontologies/IOBC`

in the layer between upper-level (fundamental) and domain-specific ontologies, providing the definition of the core concepts from a specific field. They aim at linking general concepts of a top-level ontology to more domain-specific concepts from a sub-field. Looking at the biodiversity domain, one can observe that existing ontologies tend to model parts of the domain while ignoring related parts. Furthermore, most of them connect directly to one of the existing foundational ontologies, such as BFO[7] and GFO[8]. This results in a number of challenges, e.g., the same concept can be represented in a different level of abstraction and use in different ontologies.

In this paper, we propose the design of a core ontology for the biodiversity domain using a semi-automatic approach to overcome these problems. We make use of the fusion/merge strategy [9] during the design of the core ontology, where the new ontology is developed by assembling and reusing one or more ontologies. Our design is guided by data from several databases in the biodiversity field. In particular, we develop a four-stage pipeline involving biodiversity experts and computer scientists at different phases. A set of heterogeneous biodiversity data sources is collected and analyzed. We make use of the existing ontologies from Bioportal[9] and AgroPortal[10] for extracting keywords from the collected data repository. This set of extracted terms is then filtered and revised to construct the final list of keywords. Using automated approaches of clustering and the help of biodiversity experts, we generate the list of core concepts. The links between the core concepts are discussed and determined by the domain experts.

## 2   Methodology

In this section, we describe the main steps of the proposed pipeline.

***Data Acquisition:*** The aim of this step is to get sufficient data sources from which we can extract relevant terms. To this end, we have developed a crawling method, as shown in Figure 1, considering structured and unstructured data resources. To extract relevant unstructured data, first a relaxed version of the QEMP corpus [7] is used and a number of keywords, such as *'abundance'*, *'benthic'*, *'biomass'*, *'carbon'*, *'climate change'*, *'decomposition'*, *'earthworms'*, *'ecosystem'* have been selected. The selected set of keywords is used later as input to the Semedico search engine [1] to get relevant publications from PubMed. Among them, 100 abstracts have chosen, as shown in Figure 1 reflecting the biodiversity domain by applying an iterative manual process for revision and cleaning for the crawled data. To take tabular data into consideration, we have used two well known data portals with very different characteristics (*BEFChina*[11] and

---

[7] https://bioportal.bioontology.org/ontologies/BFO
[8] https://bioportal.bioontology.org/ontologies/GFO
[9] https://bioportal.bioontology.org/
[10] http://agroportal.lirmm.fr
[11] https://china.befdata.biow.uni-leipzig.de/

*data.world*[12] ). The result of this phase is a data repository[13] which contains 100 abstracts, more than 50 tables, some datasets are given by multiple tables and, 50 metadata files.

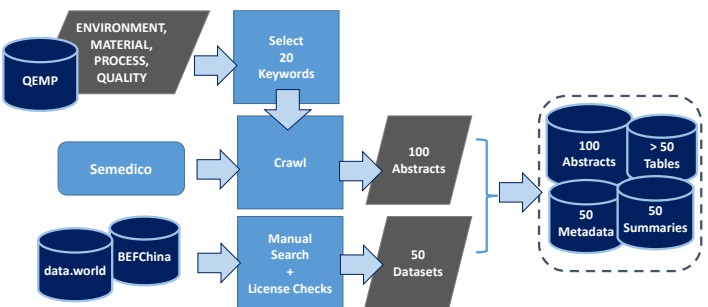

Fig. 1: Crawling phase

***Term Extraction:*** Once we have the data repository, the next step is to extract domain-specific terms. To this end, we manually annotated the collected data following the annotation scheme in [7] making use of the same ontologies and adding more important ontologies knowledge bases, like *IOBC*, *SWEET*[14], *ECOCORE*[15], *ECSO*[16], *CBO*[17], *BCO*[18] and the *Biodiversity A-Z* dictionary[19] to cover wider ranges of terms. During the extraction process, several challenges have been addressed. Our main challenge is the handling of compound words. For example, *photosynthetic O2 production* is expanded into the following keyword list: ["photosynthetic", "O2", "O2 production", "photosynthetic O2 production"]. Finally, the extracted list of terms has been enriched using other existing resources: 1) annotated keywords in QEMP corpus, 2) keywords from AquaDiva[20] project, and 3) soil related keywords [11].

***Keywords Filtration:*** To get a final list of relevant terms, we applied an automatic filtration step, where we normalized keywords to be case insensitive and in a singular form. Furthermore, we manually revised the final list of keywords to exclude spelling mistakes. At the end of this step, we have 1107 unique keywords, which is 1.8x of QEMP corpus in size and covers a broader range of Biodiversity.

---

[12] https://data.world/
[13] https://github.com/fusion-jena/BiodivOnto/tree/main/data
[14] https://bioportal.bioontology.org/ontologies/SWEET
[15] https://bioportal.bioontology.org/ontologies/ECOCORE
[16] https://bioportal.bioontology.org/ontologies/ECSO
[17] https://bioportal.bioontology.org/ontologies/CBO
[18] https://bioportal.bioontology.org/ontologies/BCO
[19] https://www.biodiversitya-z.org/
[20] http://www.aquadiva.uni-jena.de/

***Concepts and Relations Determination:*** Given the huge output list from the previous step, we have automatically calculated the intersection among our work, QEMP and AquaDiva lists. This yields a narrowed list of keywords which we define as *Seeds* as they are the most important keywords and are common among various projects dealing with Biodiversity. We have then applied a distance-based clustering technique with the objective to assign each of the remaining words to the closest seed. Seeds and words are represented by 300D word embedding using word2vec [5]. Our selected metric is the cosine similarity. Afterwards, we have manually revised the created clusters multiple times. For each revision iteration, we check how the remaining keywords are grouped, discuss the results with Biodiversity experts, and modify the selected seeds by tending to more general concepts. In the last iteration, we performed the Word-Net [8] similarity among the remaining seeds, clusters centriods, such that, if the similarity is 0.0, very unique seed, we pick this seed as a core concept. In case of having some similarity with other seeds, we have checked BioPortal for those seeds and have picked the common ancestor for them. In the previous step, we have used PATO[21], and SWEET ontologies for looking to a common ancestor. We have discussed our final list of seeds or core concepts with Biodiversity experts. Finally, we discussed the possible relations that could co-occur among our core concepts. Figure 2 represents our core categories and their core links (relations) as been validated by domain experts. Each category has a set of terms as a result of the clustering algorithm. To implement the fusion/merge strategy, we make use of the ontology modularization and selection tool (*JOYCE*) [2] to extract relevant modules from each category. Table 1 shows the results of this process. The next step is to combine (merge) the set of modules in each category to get a core ontology representing the category. All the resources related to the design of the core ontology as well as the current preliminary results are publicly available[22].

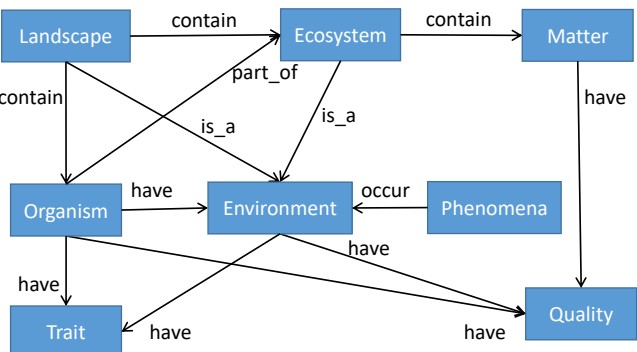

Fig. 2: Core concepts and their relations.

---

[21] https://bioportal.bioontology.org/ontologies/PATO
[22] https://github.com/fusion-jena/BiodivOnto

| Category | Ontology Modules | Terms sample inside category |
|---|---|---|
| Environment | ENVO, ECOCORE, ECSO, PATO | groundwater, garden |
| Organism | ENVO ECOCORE, ECSO, BCO | mammal, insect |
| Phenomena | ENVO, PATO, BCO | decomposition, colonization |
| Quality | ENVO, PATO, CBO, ECSO | volume, age |
| Landscape | ENVO | grassland, forest |
| Trait | BCO | texture, structure |
| Ecosystem | ENVO, ECOCORE, ECSO, PATO | biome, habitat |
| Matter | ENVO, ECSO | carbon, H2O |

Table 1: Core concepts in existing ontologies with examples.

## Acknowledgments

The authors thank the Carl Zeiss Foundation for the financial support of the project "A Virtual Werkstatt for Digitization in the Sciences (K3, P5)" within the scope of the program line "Breakthroughs: Exploring Intelligent Systems for Digitization" - explore the basics, use applications". Alsayed Algergawy' work has been funded by the *Deutsche Forschungsgemeinschaft (DFG)* as part of CRC 1076 AQUADIVA. Our sincere thanks to Tina Heger (Berlin-Brandenburg Institute of Advanced Biodiversity Research (BBIB)) as the domain expert.

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
