# OpenReview forum: "BiodivOnto: Towards a Core Ontology for Biodiversity"
_eswc-conferences.org/ESWC/2021/Conference/Poster_and_Demo_Track — ESWC2021 P&D_

### Official Review · AnonReviewer3 · 2021-04-13
**Interesting work with a clear use case**

**Rating:** 9
**Confidence:** 3

**Review:**

This paper reports the methodology and results of work for the creation of a core ontology for Biodiversity that puts together previous work in the area in a way that makes sense. The work is well motivated and the methodology followed seems sound. The results have been made available publicly in GitHub.

**Anonymity:**

Yes, I would like my review to remain anonymous.

---

### Official Review · AnonReviewer1 · 2021-04-14
**an interesting poster on the design of a core ontology for the biodiversity domain**

**Rating:** 7
**Confidence:** 4

**Review:**

This paper present presents a methodology to build a core ontology for the biodiversity domain, from relevant data and domain vocabularies and involving ontologists and domain experts.
The paper is well written and of interest for the ESWC community. All the resources to output this ontology are publicly available online (data and script).
The presented method is convincing and interesting in itself and I am wondering to which extent it may be used to design other core ontologies for other domains.
The resulting ontology is not discussed, but this can be explained by the 4 pages limit. A list of so called “seeds” is provided in the gitub repository , I was expecting its formalization in OWL with alignments with related foundational and domain ontologies.


**Anonymity:**

Yes, I would like my review to remain anonymous.

---

### Official Review · AnonReviewer4 · 2021-04-14
**Interesting approach but many important details are unclear**

**Rating:** 6
**Confidence:** 3

**Review:**

In this poster, the authors motivate the need for a core ontology for Biodiversity, and propose a method based on the fusion/merge strategy to create one. The methodology is automated and consists of data acquisition from current knowledge bases (IOBC, SWEET, Biodiversity a-z dictionary), term extraction, keyword filtering, and concepts and relations identification. The result is shown in a Figure.

The most interesting aspect of the poster is perhaps its fully automated approach to generating ontologies, which reminds of ontology learning. Specifically, the identification of concepts and relations using word embeddings and WordNet similarity on cluster centroids can be an effective way of addressing such learning in a scalable way.

However, the poster is unclear about a number of important aspects, most probably due to the limited space the authors had. In particular, the pitfalls and limitations of current ontologies in biodiversity are not discussed, and thus the what parts are currently effectively modelled and which are not is unclear. Moreover, the authors do not motivate their choice for a specific method of ontology engineering (fusion/merge); I suspect this is to address the aforementioned scalability, but this is not mentioned in the paper. Besides this, no research question is stated, and the evaluation (even in preliminary or just planned form) is missing. As minor issues, it would be good to clarify the choice of the specific data acquisition keywords, and the limitation of crawling 100 abstracts.


**Anonymity:**

Yes, I would like my review to remain anonymous.

---

### Official Review · AnonReviewer2 · 2021-04-15
**An ontology for Biodiversity mostly created by manual iterations with some assistance from clustering techniques**

**Rating:** 4
**Confidence:** 4

**Review:**

This paper describes a step-by-step process of how a core ontology is created for Biodiversity. It extracts biodiversity keywords from PubMed and other sources and applied clustering techniques using similarity scores multiple times with manual curations to generate the list of core concepts. The relationships are manually created.

Reasons to reject
- The process is mostly manual although it is assisted by the clustering techniques based on similarity scores
- Evaluation of techniques is unclear or not possible because of many manual iterations.
- The outcome is an ontology but it does not have any evaluation or comparison with other ontologies in the same domain to see the benefits or significance of this ontology.

Questions:
- Why not using pre-trained models, e.g. BERT, for generating keyword vectors?

**Anonymity:**

Yes, I would like my review to remain anonymous.

---

### Decision · Program_Chairs · 2021-04-19

Accept